**Is Khasi Society Truly Matriarchal?**

**A Critical Study of Matriliny and Gender Power in Meghalaya**

Arman Khan, Shaneya Dutta

Department of Computer Science & Engineering

University of Science & Technology Meghalaya

May 15, 2026

**Abstract**

Khasi society in Meghalaya is often described as matriarchal because descent, clan identity, and ancestral property pass through women. However, matriliny does not automatically create female domination in social or political life. This paper argues that Khasi society is better understood as matrilineal rather than matriarchal: women occupy a central position in lineage continuity and household security, but formal authority in kin mediation, village governance, and public decision-making remains largely male-dominated. Using an anthropological reading of kinship, inheritance, household roles, political institutions, and social change, the paper examines the gap between symbolic female centrality and actual gender power. The central finding is that women's inheritance rights provide status and protection, but they do not consistently translate into control over property, governance, or community norms (Nongbri, 2000; Roy, 2018).

*Keywords*: matriliny, matriarchy, Khasi society, Meghalaya, gender power, kinship, inheritance, Northeast India, customary governance, patriarchal intrusion

The Khasi of Meghalaya are among the best-known matrilineal communities in South Asia, and this has encouraged a persistent assumption that they are also matriarchal. On the surface, this assumption appears plausible: children belong to the mother's clan, husbands often live near or within the wife's family network, and the youngest daughter inherits ancestral property (Nongbri, 2013). Yet anthropology requires a more careful distinction between descent and power. A society may trace lineage through women without granting women decisive authority over property, politics, ritual, or everyday decision-making (Uberoi, 1994).

This paper takes that distinction as its central problem. It asks whether Khasi women's place in inheritance and kinship actually results in wider social and political power, or whether matriliny has been overstated as evidence of women's rule. Rather than attempting a broad survey of Khasi culture, the analysis stays focused on five linked issues: the difference between matriliny and matriarchy, the structure of Khasi inheritance, the role of male kin in authority, women's position in household and public decision-making, and the pressures of modern social change. The argument developed here is clear: Khasi society should not be classified as truly matriarchal, because women's inheritance rights are significant but domain-specific, while many decisive institutions remain under male control (Nongbri, 2000; Roy, 2018).

**Difference Between Matriliny and Matriarchy**

The distinction between matriliny and matriarchy is not merely semantic; it is the foundation of the debate. Matriliny refers to a system in which descent, clan membership, and often inheritance are traced through the mother's line. Matriarchy, in contrast, would imply that women hold primary authority in major spheres of social life, including political leadership, public decision-making, and effective control over resources. A society may therefore be matrilineal without being governed by women (Uberoi, 1994).

This distinction is especially important in anthropological analyses of the Khasi because public discussion often collapses female inheritance into female power. The stronger scholarship does not do so. Instead, it points to a recurring structural tension: women transmit the lineage, but men often mediate authority within the clan and community (Nongbri, 2013). The "matrilineal puzzle" lies precisely here — inheritance through women can coexist with male control over interpretation, dispute resolution, and institutional power (Roy, 2018).

Seen in this light, calling Khasi society "matriarchal" risks confusing symbolic centrality with governing authority. Women are indispensable to the continuity of the social order, but indispensability is not the same as dominance (Nongbri, 2000). The key analytical question is therefore not who carries the clan line, but who makes binding decisions about land, marriage, conflict, governance, and public representation (Roy, 2018).

**Khasi Kinship and Inheritance System**

Khasi kinship is organized through the kur, or matrilineal clan, and children belong to the mother's descent line rather than the father's. The best-known feature of this system is the role of the khatduh, the youngest daughter, who becomes the custodian of ancestral property and is expected to maintain the natal household (Nongbri, 2013). This arrangement gives women clear structural importance and links them directly to lineage continuity.

At the same time, the inheritance system is more limited than outsiders often assume. The khatduh is often treated not as an unrestricted owner but as a custodian charged with preserving property for the lineage (Roy, 2018). This means ancestral property cannot always be treated as an individual asset that a woman can dispose of freely. Male relatives, especially the maternal uncle, often retain influence over how such property is used and symbolically represented (Nongbri, 2013).

The role of the maternal uncle is crucial here. In classic Khasi kinship, he serves as adviser, mediator, and authority figure in clan matters. His status shows that matriliny does not eliminate male power; rather, it relocates it (Nongbri, 2000). Male authority is not centered in the father alone but is distributed through the mother's male kin, which complicates any straightforward interpretation of Khasi inheritance as female rule (Roy, 2018).

A further complication is that inheritance rights do not affect all women equally. Scholarship on Khasi women and land and forest rights shows that women's access to resources is often constrained by land alienation, livelihood change, and broader socioeconomic pressures (Shangpliang, 2012; Shangpliang, 2018). In such cases, the visibility of women in inheritance discourse can conceal inequality and the fragile material basis of women's actual autonomy.

**Gender Power and Decision-Making**

The strongest test of the matriarchy claim lies in decision-making power. If Khasi society were truly matriarchal, women's centrality in kinship would be matched by broad authority in household management, dispute resolution, and community governance. The evidence from Khasi studies points in the opposite direction (Roy, 2018).

At the household level, Khasi women bear extensive responsibility for caregiving, domestic labor, food provisioning, and in many cases agricultural and informal economic work. This labor is indispensable to family survival, but necessity does not automatically generate authority (Nongbri, 2000). Studies on Khasi family structure and gender roles show that women may be central to household continuity, yet they are not always the final decision-makers within it (Shangpliang, 2018).

This gap between responsibility and authority becomes even clearer in the public sphere. Khasi customary governance has long been organized through male-led institutions such as the dorbar shnong and related traditional councils. Women have historically been excluded from formal participation, office-holding, or meaningful voting power in these bodies (Roy, 2018). If women inherit lineage but cannot shape customary law, lead village institutions, or control public decisions, then inheritance cannot be equated with rule (Nongbri, 2013).

The political sphere is therefore where the matriarchal claim weakens most sharply. Women may be visible in markets, households, churches, and social organizations, but visibility is not the same as institutional power (Nongbri, 2000). The anthropology of Khasi society repeatedly shows a pattern of split authority: women anchor the family and lineage, while men dominate the forums in which binding collective decisions are made (Roy, 2018).

**Modern Changes and Challenges**

Modern social change has altered Khasi society, but not in a way that resolves this contradiction. Urbanization, education, Christianity, wage labor, state law, and market integration have reshaped family structure and gender expectations across Meghalaya (Nongbri, 2000). In some households, the authority of the maternal uncle has weakened, while the husband-father has become more prominent in everyday decision-making. This shift suggests that modernity has not displaced patriarchy so much as reconfigured it (Roy, 2018).

Recent debates around customary law make this even clearer. The Village Administration Bill and later lineage-related controversies showed how institutions claiming to defend Khasi tradition often preserve male authority rather than democratize it (Roy, 2018). Legal and political attempts to regulate women's marriage choices in the name of tribal identity expose a striking paradox: a society praised for female inheritance can still subject women's bodies and choices to male-controlled institutional scrutiny (Shangpliang, 2018).

Recent scholarship on domestic violence and patriarchal intrusion also complicates romantic accounts of Khasi exceptionalism. Empirical discussions from Meghalaya show that matriliny does not insulate women from abuse, control, or under-reporting of violence (Nongbri, 2013). This demonstrates that descent through women does not dissolve gender hierarchy at the level of everyday life. A matrilineal social order can still contain powerful patriarchal norms, especially when women lack equal access to institutions of law, representation, and public authority (Roy, 2018).

**Critical Analysis**

Taken together, the evidence points toward a precise conclusion: Khasi society is not truly matriarchal in the anthropological sense of women exercising primary structural power. It is matrilineal, and that matriliny matters. It gives women a recognized place in descent, secures their relationship to the natal home, and places them at the symbolic center of lineage continuity (Nongbri, 2000). These features distinguish Khasi society sharply from many patrilineal systems in South Asia (Uberoi, 1994).

But the existence of these rights should not be mistaken for gender dominance. A closer analysis shows that women's rights are often custodial rather than sovereign, while male authority persists in kin mediation, land management, village governance, and public law (Roy, 2018). Even where women influence decisions informally, formal and enforceable authority remains concentrated in male-led institutions. The major analytical mistake in popular descriptions of Khasi society is therefore the assumption that inheritance equals power (Nongbri, 2013).

A more persuasive formulation is that Khasi society combines female-centered descent with male-centered authority. This does not mean women are powerless; rather, their power is partial, relational, and unevenly distributed (Shangpliang, 2018). Some women benefit from inheritance, residence security, and social respect, while others remain constrained by land scarcity, male kin authority, and exclusion from political decision-making (Shangpliang, 2012). Once power is measured not only by who inherits but by who decides, adjudicates, represents, and governs, the label "matriarchal" becomes difficult to sustain (Roy, 2018).

**Conclusion**

Khasi society in Meghalaya should not be described as truly matriarchal. The evidence from the scholarly literature supports a more careful conclusion: Khasi society is matrilineal in descent and inheritance, but not matriarchal in the sense of women holding dominant social and political authority (Nongbri, 2000; Roy, 2018). Women's inheritance rights are meaningful because they provide continuity, status, and a degree of material security, yet those rights do not reliably translate into control over institutions, law, or collective decision-making (Shangpliang, 2012; Shangpliang, 2018).

The distinction matters for both anthropology and public debate. It prevents romanticized readings of Khasi society and allows a clearer understanding of how symbolic female centrality can coexist with durable male power (Nongbri, 2013). A critical reading of matriliny in Meghalaya therefore does not deny women's importance; it clarifies its limits. Khasi society remains one of the most important cases for showing that descent through women is not the same as rule by women (Roy, 2018).

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
