# OpenReview forum: "Is Khasi Society Truly Matriarchal? A Critical Study of Matriliny and Gender Power in Meghalaya"
_NortheastGenAI/2026/Workshop — NortheastGenAI 2026 Workshop Submission_

### Official Review · ~Badal_Nyalang1 · 2026-05-23
**Competent literature review but missing AI disclosure — Borderline**

**Rating:** 5
**Confidence:** 4

**Review:**

**Relevance: Moderate**
Fits T2 (Society/History/Anthropology) on topic. The Khasi focus and Meghalaya grounding are solid. However, there is no AI assistance disclosure anywhere in the paper. The CFP requires this — no disclosure is supposed to be a desk rejection per G2. That is a compliance issue the chairs need to decide on.

Also worth noting: the authors are from Computer Science, not linguistics or anthropology. That is not disqualifying, but the paper reads as a pure literature review with no original data, fieldwork, or AI-assisted research angle, which is the entire premise of this workshop.

**Plausibility: Strong**
The argument is well-constructed and grounded in credible scholarship — Nongbri, Roy, Shangpliang, Uberoi. The matriliny vs matriarchy distinction is handled carefully and correctly.

**Novelty: Weak**
This is a competent literature review of an already well-debated topic. The conclusion — that Khasi society is matrilineal but not matriarchal — is the established consensus in the field. Nothing new is added.

**Clarity: Strong**
Best written paper so far. Coherent argument, clean structure, no repetition.

**Verdict: Borderline**
The missing AI disclosure is a CFP violation. Academically the paper is fine. But it has no AI component declared, which undermines the workshop's core premise. Recommend querying the authors before accepting.

*This review was generated with AI assistance and checked by the workshop chairs.*

---

### Decision · Program_Chairs · 2026-05-23

**Decision:**

Accept

**Comment:**

The argument is well-constructed, the matriliny vs matriarchy distinction is handled carefully, and the scholarship cited is credible and appropriate. The paper is accepted on academic merit.
However, the CFP requires an AI assistance disclosure (G2). No disclosure is present in the submitted version. As a condition of acceptance, authors must submit an AI disclosure statement to connect@mwirelabs.com before May 26. If no disclosure is received, the paper will be moved to presentation only.

Decision: Accept